# Interpatient Heterogeneity in Drug Response and Protein Biomarker Expression of Recurrent Ovarian Cancer

**DOI:** 10.3390/cancers14092279

**Published:** 2022-05-03

**Authors:** Oliver Ingo Hoffmann, Manuel Regenauer, Bastian Czogalla, Christine Brambs, Alexander Burges, Barbara Mayer

**Affiliations:** 1SpheroTec GmbH, Am Klopferspitz 19, 82152 Martinsried, Germany; hoffmann.oliver@gmx.de; 2Department of General, Visceral and Transplant Surgery, Ludwig-Maximilians-University Munich, Marchioninistraße 15, 81377 Munich, Germany; manureg@gmx.de; 3Department of Obstetrics and Gynecology, Klinikum der Universität München, Ludwig-Maximilians-University Munich, Marchioninistraße 15, 81377 Munich, Germany; bastian.czogalla@med.uni-muenchen.de (B.C.); alexander.burges@med.uni-muenchen.de (A.B.); 4Department of Obstetrics and Gynecology, Klinikum Rechts der Isar, Technical University Munich, Ismaninger Straße 22, 81675 Munich, Germany; christine.brambs@luks.ch; 5German Cancer Consortium (DKTK), Partner Site Munich, Pettenkoferstraße 8a, 80336 Munich, Germany

**Keywords:** recurrent ovarian cancer, biomarker profiling, IHC, personalized therapy, patient-derived ovarian cancer spheroid model, 3D, tumor heterogeneity

## Abstract

**Simple Summary:**

Recurrent ovarian-cancer patients face low 5-year survival rates despite chemotherapy. Oncologists may choose from a variety of guideline-recommended second-line therapeutic options without knowing which one works best. Thus, therapy alterations and adjustments are often required. We analyzed the response that 30 tumor lesions had to certain treatments in our patient-derived ovarian-cancer spheroid model. In addition, we characterized samples by immunohistochemical staining for new druggable molecular targets. Our results might help in tailoring future therapies for individual patients with recurrent ovarian cancer.

**Abstract:**

Recurrent ovarian-cancer patients face low 5-year survival rates despite chemotherapy. A variety of guideline-recommended second-line therapies are available, but they frequently result in trial-and-error treatment. Alterations and adjustments are common in the treatment of recurrent ovarian cancer. The drug response of 30 lesions obtained from 22 relapsed ovarian cancer patients to different chemotherapeutic and molecular agents was analyzed with the patient-derived ovarian-cancer spheroid model. The profile of druggable biomarkers was immunohistochemically assessed. The second-line combination therapy of carboplatin with gemcitabine was significantly superior to the combination of carboplatin with PEGylated liposomal doxorubicin (*p* < 0.0001) or paclitaxel (*p* = 0.0007). Except for treosulfan, all nonplatinum treatments tested showed a lesser effect on tumor spheroids compared to that of platinum-based therapies. Treosulfan showed the highest efficacy of all nonplatinum agents, with significant advantage over vinorelbine (*p* < 0.0001) and topotecan (*p* < 0.0001), the next best agents. The comparative testing of a variety of treatment options in the ovarian-cancer spheroid model resulted in the identification of more effective regimens for 30% of patients compared to guideline-recommended therapies. Recurrent cancers obtained from different patients revealed profound interpatient heterogeneity in the expression pattern of druggable protein biomarkers. In contrast, different lesions obtained from the same patient revealed a similar drug response and biomarker expression profile. Biological heterogeneity observed in recurrent ovarian cancers might explain the strong differences in the clinical drug response of these patients. Preclinical drug testing and biomarker profiling in the ovarian-cancer spheroid model might help in optimizing treatment management for individual patients.

## 1. Introduction

Most patients diagnosed with primary ovarian cancer experience relapse despite extended surgery and systemic chemotherapy. Many patients diagnosed with recurrent ovarian cancer receive several lines of treatment, although each subsequent line of therapy results in shorter disease-free interval [1,2]. The prognosis of recurrent ovarian cancer is poor, documented in 5-year survival rates of less than 30% [3]. Recurrent ovarian cancer is not curable. Thus, the prolongation of progression-free survival and the maintenance of quality of life are key objectives in the treatment of recurrent ovarian cancer [4]. Numerous chemotherapeutic options and a few targeted therapies, namely, bevacizumab and PARP inhibitors, are approved for second-line treatment [5,6]. The choice of second-line therapy for individual ovarian-cancer patients depends on various factors such as treatment-free interval, previous treatments, drug-associated toxicity profiles, patient preferences, and tumor biological aspects [7]. However, the decision for the most appropriate treatment for individual patients is complex, and further selection criteria are demanded for precise oncology. The molecular heterogeneity of recurrent ovarian cancer was intensively researched [8,9,10]. In contrast, studies on heterogeneity in the protein biomarker expression and chemoresponse of recurrent ovarian cancer are rare.

The current landscape of drug response assays in ovarian cancer was reviewed in 2020 by Singh et al. [11]. This review summarizes that 3D cell culture models may be the most promising tool in preclinical research. However, three challenges remain for these models, namely, intratumor heterogeneity, intrapatient heterogeneity, and the missing tumor microenvironment.

The aim of the present study was to assess this heterogeneity in the drug response pattern in recurrent ovarian cancer. For that purpose, the ovarian-cancer spheroid model directly derived from an individual patient tumor was used for drug screening.

This 3D microtumor model closely reflects the tumor heterogeneity and architecture of the original tumor tissues [12,13], and is predictive for the drug response of the individual patient [14]. In addition, the heterogeneity of protein biomarker expression was immunohistochemically analyzed in corresponding tumor lesions to identify promising targets for molecular therapy.

## 2. Materials and Methods

### 2.1. Patients and Tumor Samples

In this prospective study, 22 patients diagnosed with recurrent epithelial ovarian carcinoma (EOC) were recruited between August 2012 and November 2014. All histological subtypes could be included. Surgical resection and pathologic diagnosis were performed according to international standard procedures [15,16,17] in the recruiting centers, i.e., the Department of Obstetrics and Gynecology, University Hospital, Ludwig Maximilians University Munich, Munich, Germany; and the Department of Obstetrics and Gynecology, Klinikum rechts der Isar, Technical University Munich, Munich, Germany. Follow-up data were assessed until September 2019.

The study was approved by the Ethics Committee of Ludwig Maximilians University, Munich, Germany (no. 278/04). Tumor samples were obtained after cancer patients had given their written informed consent. The study was performed according to the rules of good clinical practice, and certified according to ISO 9001. Surgical biospecimens were handled according to international biobanking guidelines [18]. Fresh tumor samples were used for spheroid preparation. Tumor samples snap-frozen in liquid nitrogen were used for immunohistochemistry. From 15 patients, a single tumor sample was assayed; from 6 patients, 2 tumor samples of different localizations were obtained; and from 1 patient, 3 metastatic lesions were compared.

### 2.2. Ovarian Cancer Spheroid Model and Treatment

The preparation of patient-derived cancer spheroids was described in detail [12,13,14]. In short, tumor tissue was mechanically minced and digested with an enzyme cocktail according to the manufacturer’s protocol (Roche Applied Science, Basel, Switzerland). Single-cell suspension was used for spheroid formation. Spheroids were generated in 96-well plates at 37 °C and 5% CO_2_. In each well, a single spheroid was obtained. After 48 h, spheroids were treated with a number of guideline-recommended drugs for 72 h using the peak plasma concentration (PPC) of clinically relevant doses. Each drug was tested in at least triplicate. Spheroids were exposed to both platinum- and nonplatinum-based therapies, independent from status of platinum sensitivity of the patient diagnosed with recurrent ovarian cancer. PEGylated liposomal doxorubicin (PLD) was tested on a reduced number of tumor samples because the drug was not available between 2011 and 2013 (EMA/718827/2011). Similarly, eribulin and etoposid data are not given for all tumor samples because of the limited sample size. In addition, molecular drugs erlotinib, olaparib, and vorinostat were tested, representing attractive candidates for approval [19,20]. An untreated medium control and appropriate solvent controls were included in each experiment.

The cell viability of spheroids after treatment was measured using the CellTiter-Glo^®^ Luminescent Cell Viability Assay (Promega, Fitchburg, WI, USA) according to the manufacturer’s protocol with a Tecan Ultra multiplate reader (Tecan, Männedorf, Switzerland). Counts per second (cps) values were normalized to the corresponding solvent controls. Mean viability and standard deviation were calculated for each treatment option depending on the respective solvent control.

### 2.3. Immunohistochemistry

Snap-frozen tumor tissues were embedded in tissue freezing medium (Leica Biosystems GmbH, Wetzlar, Germany). Cryoblocks were cut into 5 µm sections using a microtome (Leica Biosystems GmbH). Tissue sections were mounted onto SuperFrost slides (Thermo Scientific, Waltham, MA, USA) and air-dried overnight at room temperature. The quality of the sections, tumor morphology, and the tumor fraction were controlled by standard H&E staining.

Samples were immunohistochemically stained using the avidin–biotin–peroxidase method [21,22]. Briefly, tissue sections were fixed in acetone or for ERα detection in 4% PBS-buffered formalin for eight minutes, followed by a heat demasking process. Unspecific Fc receptors were blocked with 10% AB serum in 1 × D-PBS pH 7.4 (Bio-Rad GmbH, Dreieich, Germany) for 20 min. Endogenous biotin was blocked with a two-step avidin–biotin blocking kit (Vector Laboratories, Burlingame, CA, USA) according to the manufacturer for 20 min. Primary antibodies were applied for 1 hour. Details about the primary and secondary antibodies, and working concentrations including appropriate positive and negative controls, are given in Appendix A. Secondary biotinylated antibodies and peroxidase conjugated streptavidin (Dianova, Hamburg, Germany) were incubated for 30 min each.

The antigen–antibody reaction was visualized by incubation in 3-amino-9-ethylcarbazole pH 4.7 (Sigma-Aldrich, Steinheim, Germany) peroxidase solution for eight minutes. Tissue sections were counterstained with Mayer’s hematoxylin (Merck, Darmstadt, Germany) and embedded with Aquatex^®^ (Merck, Darmstadt, Germany). All incubation steps were carried out at room temperature and followed by three washing steps with 1 × D-PBS pH 7.4 for 10 min each.

Sections were evaluated semiquantitatively. The percentage of positively stained carcinoma cells is given for each antigen. Staining intensity was classified as weak, moderate, or strong. Her2/neu expression was scored according to breast-cancer [23] and gastric-cancer [24] guidelines. Tumors were defined as hormone receptor positive if ≥1% of the cancer cells revealed a nuclear staining of ER or PR, according to international breast-cancer guidelines [25]. PD-L1 positivity was defined as ≥1%, as described in clinical trials [26,27]. For all biomarkers, the mean expression was calculated.

### 2.4. Statistics

For statistical analysis, F-tests with Kenward–Roger approximation (with Bonferroni corrections for multiple testing and paired samples) were conducted using R version 3.2.4 (R Foundation for Statistical Computing, Vienna, Austria). Results of *p* < 0.05 were considered to be significant.

## 3. Results

### 3.1. Patient Characteristics and Clinical Treatment

A total of 30 surgically resected tumor samples obtained from 22 patients diagnosed with recurrent ovarian cancer were analyzed. Clinical and pathological characteristics of the study cohort are summarized in Table 1. Most of the EOC were diagnosed serous (86.36%). All patients were diagnosed with a high-grade (G3) tumor. A subgroup of 15 patients had a first relapse. Most of these patients (13 out of 15, 86.67%) had not received complete second-line therapy according to international standards, mostly because of comorbidities, nontolerable side effects or patient preference (Table 2). Similarly, in the patient subgroups diagnosed with a second (*n* = 3) or more frequent relapse (*n* = 4) and receiving multiple lines of chemotherapy, a number of deviations in the treatment course were documented (Table 2).

### 3.2. Tumor Sample Characteristics

The time between tumor sample removal and spheroid assay initiation in the central laboratory ranged from 14 to 24.5 h (mean 19.85 h). The amount of tumor tissue available for spheroid preparation ranged from 239.0 to 2578.7 mg (mean 1119.6 mg). Cell yield extracted from 100 mg tumor sample varied from 0.16 × 10^6^ to 4.06 × 10^6^ (mean 1.59 × 10^6^) living cells. Average cellular vitality was 91.52% (range 81.99%–96.78%).

### 3.3. Drug Response in the Ovarian Cancer Spheroid Model

In the cohort of patients diagnosed with recurrent ovarian cancer, carboplatin-based therapies, intended for platinum-sensitive relapses, revealed a reduction of the viability of ovarian cancer spheroids. Combination therapies were significantly more effective compared to carboplatin monotherapy (carboplatin combined with paclitaxel (CP) median 20.68% versus C median 28.79%, *p* < 0.0001; carboplatin combined with gemcitabine (CG) median 11.13% versus C median 28.79%, *p* < 0.0001; carboplatin combined with PEGylated liposomal doxorubicin (CPLD) median 25.49% versus C median 28.79%, *p* = 0.0002). In addition, CG was significantly more effective compared to CPLD (*p* < 0.0001) and CP (*p* = 0.0007). No significant difference was observed between CP and CPLD. Scatter plots of the platinum-based treatment results are shown in Figure 1a.

In contrast, all but one nonplatinum-based treatment option intended for platinum-resistant relapses showed a lesser effect on patient-derived cancer spheroids compared to that of carboplatin-based treatments. Treosulfan treatment resulted in strong reduction in spheroid viability that was superior to that of topotecan (treosulfan median 10.58% vs. topotecan median 41.51%, *p* < 0.0001) and vinorelbine (treosulfan median 10.58% vs. vinorelbine median 44.31, *p* < 0.0001). Treatment results of the nonplatinum-based drugs are depicted in Figure 1b. Remarkably, in five patient samples (2, 5P1, 10P1, 10P2, 15), nonplatinum treatment stimulated metabolic activity in the spheroid model.

For individual patients diagnosed with recurrent ovarian cancer of the HGSOC subtype, a differentiated drug response profile was obtained. Despite the majority of recurrences (21 out of 30, 70.0%) being primarily sensitive to CG, three tumors (2, 3, 10P2) were more vulnerable to CP. Moreover, three tumors (4, 17, 21) were equally sensitive to both combination therapies (Figure 2a). In addition, two tumors (19, 22P2) strongly responded to all three platinum-based combination treatments, namely, CP, CG, and CPLD (Figure 2b). One tumor sample (9P1) was equally responsive to carboplatin monotherapy and carboplatin combined with gemcitabine (Figure 2c). As described for individual HGSOC tumors, non-HGSOC tumors (1, 14P1 and 14P2, 20) also revealed a patient-specific rather than subtype-specific drug response profile. In summary, drug testing in the patient-derived spheroid model identified 9 out of 30 tumor samples (30%) with a different platinum response pattern than that from the result obtained in the total patient cohort, thus offering additional treatment options for individual patients. Detailed drug response results are demonstrated in Appendix A.

In 22 of 30 samples (73.33%), treosulfan had a similar (*n* = 13) or even higher (*n* = 9) effect on spheroid viability compared to the best platinum combination. For two low-platinum-sensitive tumors (1, 4), treosulfan was an effective treatment option (Figure 3). Further-line treatment reports documented the frequent modification of second-line and subsequent treatments in a number of patients resulting in short survival (Table 2). This finding supports the intensive search for further therapeutic options for individual patients. Comprehensive drug screening in platinum-resistant tumors, such as Patient 1, included not only guideline-recommended nonplatinum treatment options, but also a number of targeted therapies. Molecular drugs olaparib, erlotinib, and vorinostat, when administered as single agents, had a weak effect on spheroid viability, but this improved in combination with chemotherapy in this patient sample (Figure 4). For olaparib, the effect was only present in addition to CP, where viability was further reduced from 55.98% to 46.56%. The addition of erlotinib further reduced the cell viability of C (58.45% to 50.02%) and CP (55.98% to 41.83%), whereas vorinostat had greater impact, further reducing the cell viability of C (58.45% to 36.12%) and CP (55.98% to 31.30%).

In a few individual patients (6 and 5P1), in addition to treosulfan, ovarian-cancer spheroids responded equally well to vinorelbine and topotecan (data not shown).

#### Drug Response Pattern in Autologous Metastatic Lesions

For 7 out of 22 patients (31.82%), it was possible to analyze at least two lesions from distinctly different locations within the peritoneal cavity. All but one corresponding lesions revealed comparable response to platinum therapies. One peritoneal lesion was observed to be less platinum responsive compared to the other two corresponding locations (Patient 10). Greater disparities were found with nonplatinum treatments. In 5 out of 35 (14.29%) treatments, responses of autologous localizations differed by >20% with the same agent (Appendix A).

.

### 3.4. Biomarker Profile of Recurrent Ovarian Cancer

In 18 out of the 22 patients, metastatic tumor samples could be immunohistochemically analyzed for the expression of druggable targets (see Appendix A). In addition, two autologous lesions obtained from different peritoneal locations were characterized. The conservative approach to divide the analyzed population into a biphasic model identified a subgroup of tumors that would qualify for targeted therapy for all but one tested biomarker. Hormone receptor scoring showed a high fraction of recurrent lesions positive for ER (18 out of 23, 78.26%) and PR (13 out of 23, 56.52%). In contrast, none of the tumor samples revealed treatment-relevant (Score 3) Her2/neu protein expression. Staining for other growth factor receptors revealed a high fraction of HGF-R-positive (>mean 47.92%: 14 out of 25, 58.33%) and IGF1-R-positive (>mean 52.08%: 12 out of 24, 50.00%), but not EGF-R-positive (>mean 17.08: 6 out of 24, 25.00%) cancer cells. Staining for various cell adhesion molecules identified a strong expression of EpCAM (>mean 84.58%: 15 out of 24, 62.50%) and Muc1 (>mean 63.33%: 12 out of 24, 50.00%). In contrast, integrins α2β1 (>mean 15.87%: 5 out of 23, 21.74%) and αVβ3 (>mean 27.29%: 9 out of 24, 37.50%), and CD44v6 (>mean 23.70%: 7 out of 23, 30.43%) were detected only in a small fraction of tumor cells. Signal transduction marker rpS6 (>mean 16.67%) was expressed in 9 out of 24 samples (37.5%), and HSP90 expression (>mean 28.33%) was found in 10 out of 24 tumors (41.67%). The druggable expression of (≥1%) PD-L1 was observed in 17 out of 23 samples (73.91%). Results given in Figure 5 demonstrate a broad expression pattern of the analyzed biomarkers, indicating the biological interpatient heterogeneity of recurrent ovarian cancer independent from histological subtype. The biomarker profile obtained in non-HGSOC tumors is shown in Appendix A.

#### Biomarker Expression in Autologous Metastatic Lesions

The immunohistological comparison of 13 antigens in corresponding metastatic lesions revealed a similar biomarker expression pattern. This was most obvious for patient 22. In contrast, differences in the extent of antigen expression were observed for individual biomarkers. This was seen in patient 16 for Her2/neu and Hsp90 expression; in patient 14 for rpS6 expression; in patient 8 for EGFR expression; in patient 10 for EGFR and Hsp90 expression; and in patient 1 for HGFR, IGF1R, EpCAM and Hsp90 expression (Appendix A).

## 4. Discussion

Ovarian cancer remains a deadly disease, with 60%–80% of patients relapsing within 18 months after first-line therapy [28]. For both platinum-sensitive and -resistant patients, oncologists choose from several equally recommended chemotherapy options by major guidelines [5,6,17]. Factors relevant for drug selection are mostly age, fitness, comorbidities, and matching anticipated side effects of the prospective drug, i.e., quality of life, with an unpredictable gained lifetime. A functional in vitro biomarker platform to support decision making to maximize treatment efficiency is necessary.

In the present study, the patient-derived ovarian cancer spheroid model identified that all but one nonplatinum-based treatments were less effective than platinum-based treatments. This finding is in alignment with major guidelines, where platinum treatment is favored in platinum-sensitive recurrences. In our cohort of 22 mostly platinum-sensitive patients, the analyzed tumors significantly benefitted from platinum combination therapy in comparison to platinum used as single agent. This result is in accordance with clinical guidelines that state that a platinum combination therapy should be preferred over monotherapy in platinum-sensitive tumors [5,6]. Indeed, in the patient subgroup diagnosed with a first relapse, oncologists mostly preferred CG over CPLD. Similarly, CG was superior to CPLD and CP when comparing carboplatin combination therapies over all patients in our model. These results were confirmed in the tumors of the cohort of multiply relapsed ovarian-cancer patients, offering a treatment option where no guideline recommendation exists. Carboplatin combined with either paclitaxel or PLD had no difference in effectiveness in our test system. In a comparison study between CPLD and CP however, CPLD had an advantage in progression-free survival [29]. Our small patient cohort and the reduced number of PLD treatments might explain this discrepancy.

Only 3 out of 13 (23.08%) patients of the first relapse subgroup and two out of seven (28.57%) multiply relapsed patients received chemotherapy as recommended by the guideline, i.e., treatment was performed including full dosing of all drugs and all therapeutic cycles. In contrast, profound treatment changes were observed, which was mainly reflected by the reduction in the number of substances, dosage, and cycles as described by Eng et al. (2015) in a large-scale study with follow-up data from the 2011 TCGA study [30,31]. These studies found that treatment reduction was strongly correlated with poor prognosis for a number of patients. In the present study, this treatment heterogeneity could not be analyzed in the ovarian-cancer spheroid model, which constantly uses the PPC of the drugs given in the guideline intended dosage.

For some patients, drug testing in the ovarian-cancer spheroid model identified several more or equally effective treatment options compared to standard second-line treatment, i.e., carboplatin combined with gemcitabine. Both the exchange of one platinum-based therapy for another platinum-based therapy and the replacement of a platinum-based therapy with a nonplatinum-based therapy such as topotecan, vinorelbine and treosulfan are alternatives [32]. The choice between different treatment options seems to be most relevant for recurrent ovarian-cancer patients with comorbidities, patients experiencing side effects after the first treatment cycles, and multiply relapsed patients without any guideline options. Changing therapy might result in survival benefits, and shifting drug toxicity might improve the quality of life of individual patients. Indeed, treosulfan was identified as a safe and tolerable therapeutic option in elderly and heavily pretreated patients [33,34,35].

In addition to chemotherapeutic options, there are guideline-recommended targeted therapies, namely, bevacizumab and several PARP inhibitors. Currently, there are many attempts to identify new targets searching for new treatment options [36,37]. The present study identified strong interpatient heterogeneity in druggable protein biomarker expression, supporting the finding of molecular heterogeneity in recurrent ovarian cancer [31,38,39,40,41]. Indeed, the high ERα cut-off expression level (mean > 40% positive cancer cells) is suggested to select patients with recurrent ovarian cancer for antihormonal therapy, and might explain the low success rate in the PARAGON trial using a cut-off positivity of at least 10% for treatment [42]. Conversely, high (scores 2 and 3) Her2/neu protein expression is rare, as described by Tuefferd et al. (2007); similarly, none of the patients in the present study qualified for anti-Her2/neu treatment [43]. Most recurrent ovarian cancers in the small subgroup of seven patients did not differ in protein biomarker expression on an intrapatient level. Some ovarian-cancer clinical trials use this approach to stratify their study population, excluding patients with histologically low expression of the druggable biomarker of the corresponding targeted therapy (e.g., NCT04460807, NCT03078400). In contrast, most clinical trials do not use protein biomarkers for patient stratification hindered by high costs [44].

Molecular pathology studies of primary high-grade serous ovarian carcinomas found that, despite histologically uniform morphology, there is a high degree of intratumor and interpatient heterogeneity in these cancers [45]. Studies found an underlying branching process into different molecular tumor areas in the primary tumor and the independent adaptation of tumor lesions to the respective environment at different sites, while maintaining the morphological subtype [46,47]. The genomic landscape of primary ovarian cancer is complex and highly dynamic. Regional, spatial, and temporal heterogeneity is also influenced by numerous intrinsic and extrinsic factors such as chromosomal instability [48], the tumor microenvironment, and chemotherapy. Current data suggest that selection pressure from these interferences may lead to the emergence of pre-existing resistant clones rather than de novo mutations in recurrent disease [49]. These different genomic profiles are associated with treatment response and survival [50].

Similarly, our study of recurrent ovarian carcinomas showed that there is a striking heterogeneity of biomarker expression, and a variety of different druggable targets were identified between patients. Supporting results were obtained for the functional level, i.e., the ovarian-cancer spheroid model showing differences in response profiles between patients. This phenotypic and functional heterogeneity was observed in both the majority of HGSOC tumors and non-HGSOC tumors, although the number of this subtype was small.

Contrary to the described situation in primary tumors, there were rarely differences on the protein level between different metastatic lesions of the same recurrent patient. Already administered treatment (after excision of the previous tumor) may have exerted uniform evolutionary pressure on the remaining cells. Thus, a single biopsy of the primary tumor most probably does not represent the entire tumor; most importantly, relapses may express targetable molecular changes that are not present in the original tumor [51]. This supports the request to routinely assess treatment-related biomarkers in newly diagnosed metastatic lesions before treatment, as already recommended for ER, PR, Her2/neu, and Ki67 or PD-L1 in recurrent mamma carcinoma [52]. A number of approved molecular drugs in nonovarian cancers could represent a tailored treatment option for ovarian-cancer patients as off-label therapy. Indeed, examples for clinical studies are available for ganetespib directed against Hsp-90 [53], and atezolizumab directed against PD-L1 [54], with some trials still ongoing (e.g., NCT04931342, NCT03598270).

In summary, the heterogeneous biology of ovarian cancer requires a detailed profiling of the individual tumor, resulting in specific drug selection for individual ovarian-cancer patients. These molecular treatment candidates and guideline-recommended drugs for platinum-sensitive and platinum-resistant tumors may be tested side by side in the functional patient-specific ovarian-cancer spheroid model. In future studies, the predictivity of the ovarian-cancer spheroid model must be confirmed in a large patient cohort.

## 5. Conclusions

Recurrent ovarian cancer was characterized by high variability in druggable target expression and drug response profiling in our cohort of patients. This interpatient tumor heterogeneity was modeled on the patient-derived ovarian-cancer spheroid model. Preclinical drug selection using this functional approach may lead to improved survival of individual ovarian-cancer patients.

## Figures and Tables

**Figure 1 cancers-14-02279-f001:**
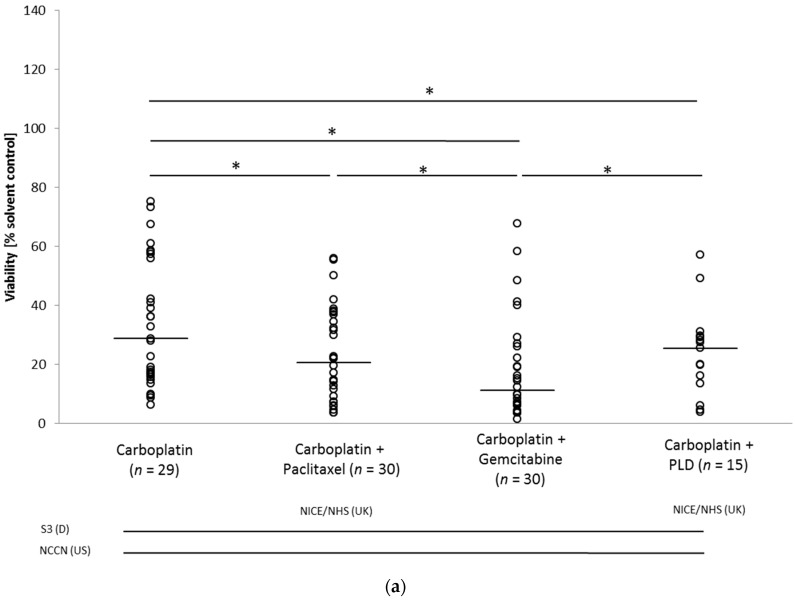
Scatter plot of chemotherapeutic response obtained in patient-derived ovarian cancer spheroid model of total patient cohort against standard (**a**) platinum- and (**b**) nonplatinum-based therapies. Dots, tumor samples. Short bars, median cell viability for each treatment option. Long bars, treatment recommendations in different national and international guidelines at time of sample recruitment. PLD, PEGylated liposomal doxorubicin; * *p* < 0.05.

**Figure 2 cancers-14-02279-f002:**
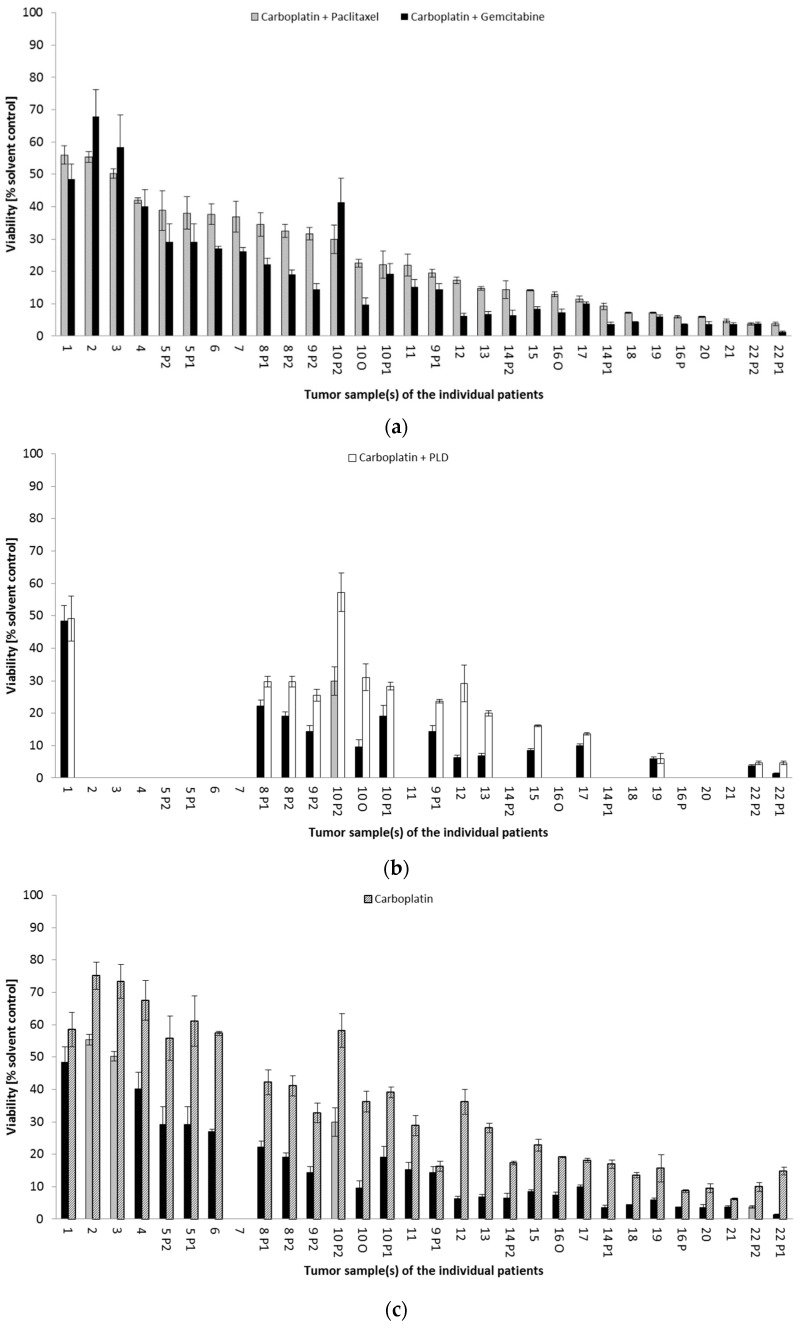
Comparison of different platinum-based treatment options in patient-derived ovarian-cancer spheroid model of individual patients. Each bar represents a treatment option. (**a**) Carboplatin + paclitaxel compared to carboplatin + gemcitabine; (**b**) best therapy from (**a**) compared to carboplatin + PLD; (**c**) best therapy from (**a**,**b**) compared to carboplatin monotherapy. Each number on *x*-axis represents an individual patient. In seven patients (5, 8, 9, 10, 14, 16, 22), tumor samples from different locations were comparatively analyzed: O, omental lesion; P1, peritoneal lesion 1; P2, peritoneal lesion 2. Gaps indicate that not all treatment options could be tested in all tumor samples as described in Section 2.

**Figure 3 cancers-14-02279-f003:**
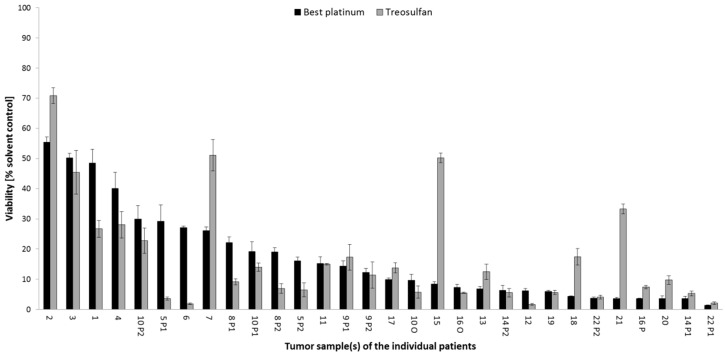
Comparison of best platinum-based therapy (from Figure 2) with treosulfan treatment in spheroid model. Further explanations given in Figure 2.

**Figure 4 cancers-14-02279-f004:**
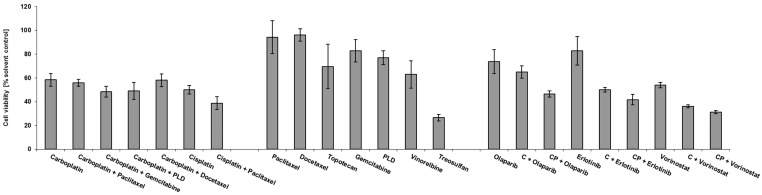
Drug response profile in spheroid model prepared from pararectal peritoneal metastasis of individual patient 1. PLD, PEGylated liposomal doxorubicin; C, carboplatin; CP, carboplatin combined with paclitaxel.

**Figure 5 cancers-14-02279-f005:**
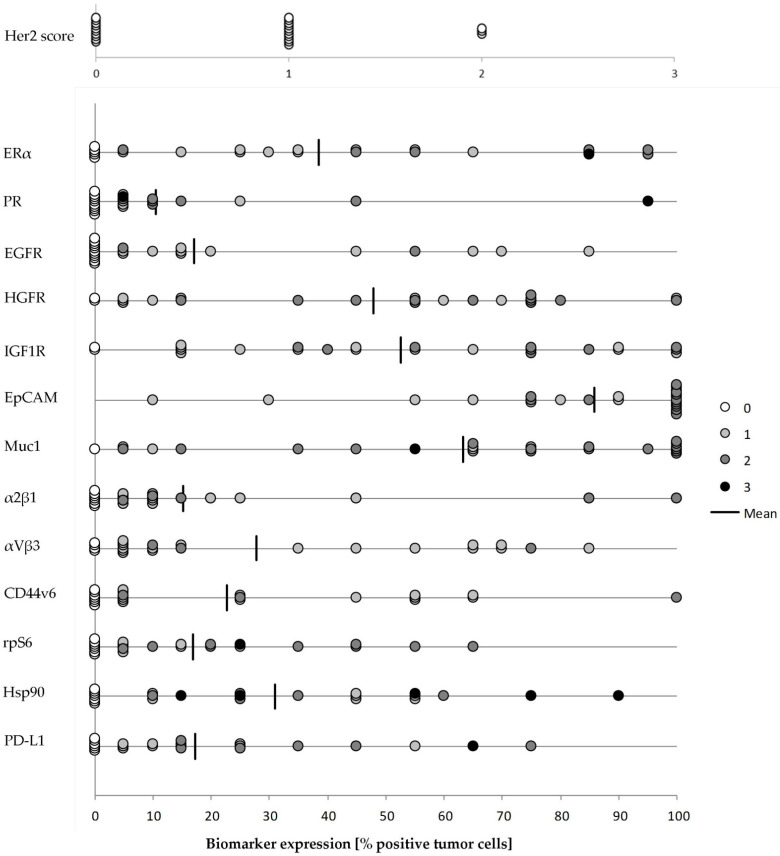
Immunohistochemical biomarker profiling in recurrent ovarian cancer. Each dot represents a tumor sample. Gray scale intensity represents staining intensity (0 = negative; 1 = weakly positive; 2 = moderately positive; 3 = strongly positive). Evaluation criteria described in Section 2.

**Table 1 cancers-14-02279-t001:** Patient characteristics.

Characteristic	No.	%
Age, years		
Mean/median	60/61	
Range	37–76	
Relapse number		
1	15	68.18
2	3	13.64
>2	4	18.18
Histology		
Serous	19	86.36
Other	3	13.64
Grading		
Grade 3	22	100
Macroscopic residual tumor after surgery of the current relapse		
Yes	9	40.91
No	13	59.09
Number of analyzed samples per patient		
1	15	68.18
2	6	27.27
3	1	4.55

**Table 2 cancers-14-02279-t002:** Treatment and follow-up of patients diagnosed with first relapse and with subsequent relapse.

**Relapse** **Number**	**Patient No.**	**PFI (mo)**	**Second-Line** **Chemotherapy**	**PFS (mo)**	**OS (mo)**	**Tumor-Related Death**
1	3	13	CP + B	17	47	-
	4	20	C(AUC4)G(800 mg/m^2^) -> B	30	33	Yes
	6	22	2 × C(AUC2)G, 4 × C(AUC1)D(35 mg/m^2^)	25	36	Yes
	9	11	CPLD (80%Cy3–6)	7	57	-
	10	1	4 × CP, 2 × CP(75%)	4	11	Yes
	11	8	C(AUC4)G + B(75%Cy2,3 Ød8Cy4,5)	22	32	Yes
	12	15	C(AUC4)G + B	17	51	-
	13	23	1 × CPLD, 5 × CG	43	75	-
	17	67	2 × CPLD(85%), 2 × C(AUC5), 2 × C(AUC4)	12	43	-
	18	11	CPLD + B	27	31	Yes
	21	13	C(AUC4)G (Ød8Cy2,3,6)	8	41	Yes
	22	59	C(AUC4)G + B(red. Cy1,2,6 d8Cy5,6)	17	71	-
	20	4	1 × G, 5 × T(3 mg/m^2^), 14 × T(4 mg/m^2^)	16	18	Yes
	5	13	None	5	11	Yes
	19	262	None	61	61	-
**Relapse Number**	**Patient No.**	**TFI (mo)**	**Subsequent** **Chemotherapy**	**PFS** **(mo)**	**OS** **(mo)**	**Tumor-Related Death**
2	16	13	3 × CG, 3 × C	17	77	-
2	15	15	CP + B	9	29	Yes
3	14	8	4 × C(AUC4)G(800 mg/m^2^), 2 × Cis(50 mg/m^2^)G(800 mg/m^2^)	6	13	Yes
3	8	59	2 × C, 6 × PLD	65	65	-
5	7	5	Cis(20 mg/m^2^)D(30 mg/m^2^) + B(Ød15Cy3, aborted Cy6)	8	9	Yes
2	1	1	9 × T(1.25 mg/m^2^)	54	54	-
4	2	1	None	2	2	Yes

Patients grouped into platinum-based therapy, nonplatinum-based therapy, and no therapy, followed by patient number. PFI, platinum-free interval; TFI, treatment-free interval (time between end of last therapy and subsequent relapse); PFS, progression-free survival; OS, overall survival; Cy, treatment cycle; d, day; mo, months; subsequent chemotherapy, administered treatment for examined relapse; -, alive at last follow-up; brackets, deviation from guideline dosing regimen; red., reduced; ->, changed treatment; Ø, skipped day of treatment. Drug abbreviations: B, bevacizumab; C, carboplatin; Cis, cisplatin; D, docetaxel; G, gemcitabine; P, paclitaxel; PLD, PEGylated liposomal doxorubicin; T, topotecan.

## Data Availability

The data presented in this study are available on request from the corresponding author.

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
