# Peer review of "Interpatient Heterogeneity in Drug Response and Protein Biomarker Expression of Recurrent Ovarian Cancer"

_cancers, 2022, doi:10.3390/cancers14092279_

Round 1

Reviewer 1 Report

Thank you for allowing me to review the manuscript entitled „Inter-patient heterogeneity in drug response and protein bi- 2 omarker expression of recurrent ovarian cancer” by Oliver Ingo Hoffmann, Manuel Regenauer, Bastian Czogalla, Christine Brambs, Alexander Burges and Barbara Mayer.

This is a prospective study including 22 patients with recurrent ovarian cancer aiming to asses the heterogeneity in drug response pattern. An ovarian cancer spheroid model derived from individual patient tumor was used to avoid limitations from conventional tumor cell lines.

The study addresses an important topic on a common malignant disease with high mortality rates.

There are however a few limitations and issues that should be addressed:

As the authors have nicely elucidated, molecular studies on primary high-grade serous ovarian cancer (HGSOC) have shown that there is a high degree of intra-tumor and inter-patient heterogeneity.

As such, HGSOC is a disease in itself that demonstrates a high degree of variability. In order to draw any meaningful conclusions, the authors should limit their samples to HGSOC (i.e. remove the 3 non-serous histologies). In addition, the authors should confirm that all serous cancers were HGSOC. It is well known that low-grade serous ovarian cancer (LGSOC) is a different disease entity with different tumor biology. Thus, any LGSOC in this cohort should also be removed.

Otherwise, it remains unclear, which patient would benefit from which therapies. The authors report for instance that 3 patient tumor samples had better response to CP than CG (were these HGSOC or other histologies)?

It is also interesting that there was little to no effect from Olaparib. This is in contrast to clinical studies. How do the authors explain this discrepancies?

Why was the cut-off for follow-up chosen back in 2019? Are there any updates?

Did the statistical model account for repeated testing?

Author Response

Manuscript ID cancers-1667043

Inter-patient heterogeneity in drug response and protein biomarker expression of recurrent ovarian cancer

Reply to Reviewer 1

Moderate English changes required

Answer:

We have proofread sentence by sentence again for English. However, the original manuscript was extensively proofread for grammar and general English by Leslie D. Patterson, a native speaker and PhD at the University of Notre Dame in Indianapolis, USA. We have given her special thanks in the Acknowledgments for her linguistic revision.

Remark 1:

“As the authors have nicely elucidated, molecular studies on primary high-grade serous ovarian cancer (HGSOC) have shown that there is a high degree of intra-tumor and inter-patient heterogeneity. As such, HGSOC is a disease in itself that demonstrates a high degree of variability. In order to draw any meaningful conclusions, the authors should limit their samples to HGSOC (i.e. remove the 3 non-serous histologies). In addition, the authors should confirm that all serous cancers were HGSOC. It is well known that low-grade serous ovarian cancer (LGSOC) is a different disease entity with different tumor biology. Thus, any LGSOC in this cohort should also be removed. Otherwise, it remains unclear, which patient would benefit from which therapies. The authors report for instance that 3 patient tumor samples had better response to CP than CG (were these HGSOC or other histologies)?”

Answer 1:

Thank you very much for this valuable feedback. Table 1 (Patient characteristics) shows that all analyzed lesions were diagnosed as high grade (grade 3). No LGSOC were included in the present study.

Although numerous publications point out the biological heterogeneity of the histologic subtypes of ovarian cancer, to date this has no therapeutic consequences according to the current international guidelines. Thus, all histologic subtypes receive combination therapy of carboplatin and paclitaxel. Accordingly, we did not differentiate by histologic subtype in the present study and described a patient cohort representative of the mixed clinical population including both serous and non-serous subtypes.

The histology of the three tumor samples (patients 1, 14p1 and 14P2, 20) grouped under "Other" was adenocarcinoma (n=1) and undifferentiated adenocarcinoma (n=2). The response of these patients to the drug was quite variable (Fig. 2, Fig. 3). Although patient samples 20, 14P1, and 14P2 were diagnosed non-serous carcinomas, they were found to be chemosensitive to platinum-based therapy as described for HGSOC. In contrast, drug testing of patient sample 1 revealed poor response to platinum-containing, non-platinum-containing, and targeted therapies (Fig. 2, Fig. 3, and Fig. 4). This observation suggests that drug testing should be performed regardless of histologic subtype to select the most effective treatment for the individual patient.

Remark 2:

“It is also interesting that there was little to no effect from Olaparib. This is in contrast to clinical studies. How do the authors explain this discrepancies?”

Answer 2:

We tested olaparib in only one patient sample (Pat 1). The frequency of BRCA mutations/HRD ranges from 11 to 16% (Jasiewicz et al, 2022; Bookman et al, 2017 and Risch et al, 2006). It is very possible that Pat 1 did not have a BRCA mutation and therefore no effect of olaparib treatment was observed. In addition, a number of mechanisms of acquired PARP inhibitor resistance have been described, such as restoration of HRR due to restoration of BRCA1/2 function, restoration of replication fork stability, or mutations in PARP or functionally related proteins ( reviewed in Tao and Wu, 2021; Jiang et al, 2018). In addition, upregulation of drug efflux pumps may play a specific role for Pat 1's tumor, which was resistant not only to olaparib but to several different drugs.

Remark 3:

“Why was the cut-off for follow-up chosen back in 2019? Are there any updates?”

Answer 3:

The endpoint of the study was 5-year survival, which was approved by the ethics committee. We chose 5-year survival because it is a common endpoint in ovarian cancer trials. After enrolling the last patient in 2014, the planned follow-up ended in 2019. This is described in the original manuscript (2.1 Patients and Tumor Samples, lines 79-88).

Remark 4:

“Did the statistical model account for repeated testing?”

Answer 4:

We state in 2.4. Statistics that F-tests with Kenward-Roger approximation with corrections for multiple testing and paired samples were conducted (line 151-152). To be more precise we added the type of correction in the manuscript (line 151).

Reviewer 2 Report

In this paper, Authors show the analysis of the drug response, of 30 lesions obtained from 22 relapsed ovarian cancer patients, to different chemotherapeutic and molecular agents with the patient-derived ovarian cancer spheroid model. They tested drug treatment against sferoid from tumour cells and the profile of drugable biomarkers was assessed immunohistochemically.

By means of these methods they observed a biological heterogeneity observed in recurrent ovarian cancers and conclude that it might explain the strong differences in the clinical drug response of these patients and in addition, suggest that these analysis in the ovarian cancer spheroid model might help to optimize treatment management for the individual patient.

The work seems well done even if on a not so large number of patients.

Furthermore, I also wonder if the heterogeneity observed may also be due to multiple lines of chemotherapies.

Another concern could be that evaluation has been done by means of immunohistochemistry alone, of which, however, there are no images of even some analyzes not even in the supplementaries.

Perhaps it would not be appropriate to verify some analyzes also by means of Western blot?

Author Response

Manuscript ID cancers-1667043

Inter-patient heterogeneity in drug response and protein biomarker expression of recurrent ovarian cancer

Reply to Reviewer 2

Remark 1:

“Furthermore, I also wonder if the heterogeneity observed may also be due to multiple lines of chemotherapies.”

Answer 1:

Thank you for this interesting question. We have expanded the discussion in the original manuscript (lines 375-381) on the heterogeneity of primary and recurrent ovarian cancer and included intrinsic and extrinsic factors, such as chemotherapy, that might lead to the emergence of pre-existing resistant clones rather than de novo mutations. New literature was added.

Remark 2:

“Another concern could be that evaluation has been done by means of immunohistochemistry alone, of which, however, there are no images of even some analyzes not even in the supplementaries.”

Answer 2:

To address this concern, we have included the new Figure S3 with images of IHC staining in the supplemental section.

Remark 3:

“Perhaps it would not be appropriate to verify some analyzes also by means of Western blot?”

Answer 3:

Unfortunately, we cannot implement this interesting suggestion. There are several reasons for this: 1) the sample size allocated to us by the pathology was mostly limited and used completely for the studies presented. 2) We received the samples anonymized as part of the study. Therefore, it is difficult for us to establish a reference to the non-coded samples, which might be stored in the pathology. 3) Even if we receive the FFPE samples from pathology through a broker, the quality of the samples could be compromised after a storage period of 8 to 10 years.

Round 2

Reviewer 1 Report

Again thank you for allowing me to review the manuscript entitled: "Inter-patient heterogeneity in drug response and protein biomarker expression of recurrent ovarian cancer by Oliver Ingo Hoffmann , Manuel Regenauer , Bastian Czogalla , Christine Brambs , Alexander Burges and Barbara Mayer.

The authors have now provided a revised version. They have not considered to limit their analysis to HGSOC, which I still think is a flaw. As the authors have pointed out in their reply the current treatment approach is "a one-size fits all" and by again combining all histologies in this study we will not get any closer to what is commonly referred to "precision medicine". 

Author Response

Manuscript ID cancers-1667043

Inter-patient heterogeneity in drug response and protein biomarker expression of recurrent ovarian cancer

Reply to Reviewer 1 / Round 2

Reviewers Point:

“The authors have now provided a revised version. They have not considered to limit their analysis to HGSOC, which I still think is a flaw. As the authors have pointed out in their reply the current treatment approach is "a one-size fits all" and by again combining all histologies in this study we will not get any closer to what is commonly referred to "precision medicine". 

Answer:

We fully agree with Reviewer 1 that HGSOC represents a heterogeneous disease. In addition, we fully agree with the reviewer that "one-size fits all" is outdated, which is why we have presented tumor heterogeneity in recurrent ovarian cancer phenotypically and functionally. However, the present prospective study (Ethics approval, No.278/04) was not intended to focus on the HGSOC. All histological subtypes of recurrent ovarian cancer could be included, similar as described in other current studies (see publication list below). The main reason was that precision diagnostics should be used for each individual patient regardless of the tumor subtype. And indeed, the three non-HGSOC tumors analysed in the present study showed heterogeneous results in both protein signature and drug response profile similar as observed for the HGSOC tumors.

We have found the following solution and hope that we will meet both the concern of the reviewer and the study design:

1) We maintained the three non-HGSOC (“others”) and described the study design in 2.1 Patients and tumor samples (lines 78 and 79) accordingly.

2) We differentiated the results obtained for HGSOC and non-HGSOC for the drug response profile (3.3, lines 211-212 and lines 219-221) and the biomarker profile (3.4, line 289-290; a new Figure S4 depicting the non-HGSOC tumors only is given).

3) We discussed tumor heterogeneity for both HGSOC and non-HGSOC (4. Discussion, lines 386-388).

We thank the Reviewer for the valuable contribution, which resulted in a more differentiated consideration of HGSOC and non-HGSOC patients. This has substantially improved our manuscript.

Publications (examples) considering both HGSOC and non-HGSOC within the same study:

  1. Adam, J.-P.; Boumedien, F.; Letarte, N.; Provencher, D. Single Agent Trabectedin in Heavily Pretreated Patients with Recurrent Ovarian Cancer. Gynecologic Oncology 2017, 147, 47–53, doi:10.1016/j.ygyno.2017.07.123.
  2. Chen, W.; Ye, S.; Wu, Y.; Pei, X.; Xiang, L.; Ping, B.; Shan, B.; Yang, H. Changes in Peripheral Lymphocyte Populations in Patients with Advanced/Recurrent Ovarian Cancer Undergoing Splenectomy during Cytoreductive Surgery. J Ovarian Res 2021, 14, 113, doi:10.1186/s13048-021-00860-7.
  3. Costales, A.B.; Chambers, L.; Chichura, A.; Rose, P.G.; Mahdi, H.; Michener, C.M.; Yao, M.; Debernardo, R. Effect of Platinum Sensitivity on the Efficacy of Hyperthermic Intraperitoneal Chemotherapy (HIPEC) in Recurrent Epithelial Ovarian Cancer. Journal of Gynecology Obstetrics and Human Reproduction 2021, 50, 101844, doi:10.1016/j.jogoh.2020.101844.
  4. Coward, J.I.; Barve, M.A.; Kichenadasse, G.; Moore, K.N.; Harnett, P.R.; Berg, D.; Garner, J.S.; Dizon, D.S. Maximum Tolerated Dose and Anti-Tumor Activity of Intraperitoneal Cantrixil (TRX-E-002-1) in Patients with Persistent or Recurrent Ovarian Cancer, Fallopian Tube Cancer, or Primary Peritoneal Cancer: Phase I Study Results. Cancers 2021, 13, 3196, doi:10.3390/cancers13133196.
  5. Gonçalves Ribeiro, A.R.; Marineli Salvadori, M.; de Brot, L.; Bovolin, G.; Mantoan, H.; Ilelis, F.; Rezende, M.; Soares do Amaral, N.; Moraes Sanches, S.; Lisboa Maya, J.M.; et al. Retrospective Analysis of the Role of Cyclin E1 Overexpression as a Predictive Marker for the Efficacy of Bevacizumab in Platinum-Sensitive Recurrent Ovarian Cancer. ecancer 2021, 15, doi:10.3332/ecancer.2021.1262.
  6. Hsu, C.-C.; Pan, Y.-B.; Lai, C.-H.; Chang, T.-C.; Yang, L.-Y.; Chou, H.-H. Olaparib as Maintenance Therapy and Salvage Therapy in Recurrent Ovarian Cancer: The Early Experience in Taiwan. Taiwanese Journal of Obstetrics and Gynecology 2021, 60, 634–638, doi:10.1016/j.tjog.2021.05.010.
  7. Izuchi, D.; Fukagawa, S.; Yotsumoto, F.; Shigekawa, K.; Yoshikawa, K.; Hirakawa, T.; Kiyoshima, C.; Ouk, N.S.; Urushiyama, D.; Katsuda, T.; et al. Association of Serum HB-EGF Value and Response to Chemotherapy in Patients with Recurrent Ovarian Cancer. Anticancer Res 2018, 38, 4347–4351, doi:10.21873/anticanres.12735.
  8. Kim, J.H.; Ha, H.I.; Kim, M.H.; Han, M.R.; Park, S.-Y.; Lim, M.C. A Modified Intraperitoneal Chemotherapy Regimen for Ovarian Cancer: Technique and Treatment Outcomes. Cancers 2021, 13, 4886, doi:10.3390/cancers13194886.
  9. Lee, J.-Y.; Kim, B.-G.; Kim, J.-W.; Lee, J.B.; Park, E.; Joung, J.-G.; Kim, S.; Choi, C.H.; Kim, H.S.; on behalf of Korean Gynecologic Oncology Group (KGOG) investigators Biomarker-Guided Targeted Therapy in Platinum-Resistant Ovarian Cancer (AMBITION; KGOG 3045): A Multicentre, Open-Label, Five-Arm, Uncontrolled, Umbrella Trial. J Gynecol Oncol 2022, 33, e45, doi:10.3802/jgo.2022.33.e45.
  10. Lee, M.-W.; Ryu, H.; Song, I.-C.; Yun, H.-J.; Jo, D.-Y.; Ko, Y.B.; Lee, H.-J. Efficacy of Cisplatin Combined with Topotecan in Patients with Advanced or Recurrent Ovarian Cancer as Second- or Higher-Line Palliative Chemotherapy. Medicine 2020, 99, e19931, doi:10.1097/MD.0000000000019931.
  11. Li, C.; Bonazzoli, E.; Bellone, S.; Choi, J.; Dong, W.; Menderes, G.; Altwerger, G.; Han, C.; Manzano, A.; Bianchi, A.; et al. Mutational Landscape of Primary, Metastatic, and Recurrent Ovarian Cancer Reveals c-MYC Gains as Potential Target for BET Inhibitors. Proc Natl Acad Sci USA 2019, 116, 619–624, doi:10.1073/pnas.1814027116.
  12. Liao, J.B.; Gwin, W.R.; Urban, R.R.; Hitchcock-Bernhardt, K.M.; Coveler, A.L.; Higgins, D.M.; Childs, J.S.; Shakalia, H.N.; Swensen, R.E.; Stanton, S.E.; et al. Pembrolizumab with Low-Dose Carboplatin for Recurrent Platinum-Resistant Ovarian, Fallopian Tube, and Primary Peritoneal Cancer: Survival and Immune Correlates. J Immunother Cancer 2021, 9, e003122, doi:10.1136/jitc-2021-003122.
  13. Nakanishi, K.; Yamada, T.; Ishikawa, G.; Suzuki, S. Beyond BRCA Status: Clinical Biomarkers May Predict Therapeutic Effects of Olaparib in Platinum-Sensitive Ovarian Cancer Recurrence. Front. Oncol. 2021, 11, 697952, doi:10.3389/fonc.2021.697952.
  14. Narui, C.; Tanabe, H.; Shapiro, J.S.; Nagayoshi, Y.; Maruta, T.; Inoue, M.; Hirata, Y.; Komazaki, H.; Takano, H.; Niimi, S.; et al. Readministration of Platinum Agents in Recurrent Ovarian Cancer Patients Who Developed Hypersensitivity Reactions to Carboplatin. In Vivo 2019, 33, 2045–2050, doi:10.21873/invivo.11702.
  15. Raj, H.; Santosh Keerthi, M.S.; Palaniappan, R.; Prakash, U.; Dhanushkodi, M.; Ganesan, T.S. Phase 2 Non-Randomised Trial of Secondary Cytoreduction and Hyperthermic Intraperitoneal Chemotherapy in Recurrent Platinum-Sensitive Ovarian Cancer. ecancer 2021, 15, doi:10.3332/ecancer.2021.1260.
  16. Trédan, O.; Provansal, M.; Abdeddaim, C.; Lardy-Cleaud, A.; Hardy-Bessard, A.-C.; Kalbacher, E.; Floquet, A.; Venat-Bouvet, L.; Lortholary, A.; Pop, O.; et al. Regorafenib or Tamoxifen for Platinum-Sensitive Recurrent Ovarian Cancer with Rising CA125 and No Evidence of Clinical or RECIST Progression: A GINECO Randomized Phase II Trial (REGOVAR). Gynecologic Oncology 2022, 164, 18–26, doi:10.1016/j.ygyno.2021.09.024.
  17. Wang, C.-W.; Lee, Y.-C.; Chang, C.-C.; Lin, Y.-J.; Liou, Y.-A.; Hsu, P.-C.; Chang, C.-C.; Sai, A.-K.-O.; Wang, C.-H.; Chao, T.-K. A Weakly Supervised Deep Learning Method for Guiding Ovarian Cancer Treatment and Identifying an Effective Biomarker. Cancers 2022, 14, 1651, doi:10.3390/cancers14071651.
  18. Xia, L.; Peng, J.; Lou, G.; Pan, M.; Zhou, Q.; Hu, W.; Shi, H.; Wang, L.; Gao, Y.; Zhu, J.; et al. Antitumor Activity and Safety of Camrelizumab plus Famitinib in Patients with Platinum-Resistant Recurrent Ovarian Cancer: Results from an Open-Label, Multicenter Phase 2 Basket Study. J Immunother Cancer 2022, 10, e003831, doi:10.1136/jitc-2021-003831.
  19. Xu, J.; Gao, Y.; Luan, X.; Li, K.; Wang, J.; Dai, Y.; Kang, M.; Lu, C.; Zhang, M.; Lu, C.X.; et al. An Effective AKT Inhibitor-PARP Inhibitor Combination Therapy for Recurrent Ovarian Cancer. Cancer Chemother Pharmacol 2022, doi:10.1007/s00280-022-04403-9.
  20. Yoshida, H.; Shintani, D.; Fujiwara, K. Obesity Is a Predictive Biomarker of Poor Benefit from Single-Agent Bevacizumab Therapy in Recurrent Ovarian Cancer Patients. J BUON 2021, 26, 1762–1767.
  21. D’Alterio, C.; Spina, A.; Arenare, L.; Chiodini, P.; Napolitano, M.; Galdiero, F.; Portella, L.; Simeon, V.; Signoriello, S.; Raspagliesi, F.; et al. Biological Role of Tumor/Stromal CXCR4-CXCL12-CXCR7 in MITO16A/MaNGO-OV2 Advanced Ovarian Cancer Patients. Cancers 2022, 14, 1849, doi:10.3390/cancers14071849.
  22. De Bruyn, C.; Ceusters, J.; Landolfo, C.; Baert, T.; Thirion, G.; Claes, S.; Vankerckhoven, A.; Wouters, R.; Schols, D.; Timmerman, D.; et al. Neo-Adjuvant Chemotherapy Reduces, and Surgery Increases Immunosuppression in First-Line Treatment for Ovarian Cancer. Cancers 2021, 13, 5899, doi:10.3390/cancers13235899.
  23. Gyllensten, U.; Hedlund-Lindberg, J.; Svensson, J.; Manninen, J.; Öst, T.; Ramsell, J.; Åslin, M.; Ivansson, E.; Lomnytska, M.; Lycke, M.; et al. Next Generation Plasma Proteomics Identifies High-Precision Biomarker Candidates for Ovarian Cancer. Cancers 2022, 14, 1757, doi:10.3390/cancers14071757.
  24. Lalos, A.; Neri, O.; Ercan, C.; Wilhelm, A.; Staubli, S.; Posabella, A.; Weixler, B.; Terracciano, L.; Piscuoglio, S.; Stadlmann, S.; et al. High Density of CD16+ Tumor-Infiltrating Immune Cells in Recurrent Ovarian Cancer Is Associated with Enhanced Responsiveness to Chemotherapy and Prolonged Overall Survival. Cancers 2021, 13, 5783, doi:10.3390/cancers13225783.

Round 3

Reviewer 1 Report

The authors have addressed all comments and suggestions.